# Non-invasive myocardial work is reduced during transient acute coronary occlusion

Jolanda Sabatino[1,2,3], Salvatore De Rosa[1,2]*, Isabella Leo[1], Carmen Spaccarotella[1,2], Annalisa Mongiardo[1], Alberto Polimeni[1,2], Sabato Sorrentino[1,2], Giovanni Di Salvo[3,4], Ciro Indolfi[1,2,5]*

1 Division of Cardiology, Department of Medical and Surgical Sciences, "Magna Graecia" University, Catanzaro (CZ), Italy, 2 Cardiovascular Research Center, "Magna Graecia" University, Catanzaro (CZ), Italy, 3 Department of Paediatric Cardiology, Royal Brompton Hospital, London, United Kingdom, 4 Department of Women's and Children's Health, University of Padua, Padua, Italy, 5 Mediterranea Cardiocentro, Naples (NA), Italy

* saderosa@unicz.it (SDR); indolfi@unicz.it (CI)

**Data Availability Statement:** The data underlying the results presented in the study are available from Open Science Framework: https://osf.io/bqnmh/.

## Abstract

### Background

During ischemia a close relationship exists between sub-endocardial blood flow and myocardial function. Strain parameters can capture an impairment of regional longitudinal function but are load dependent. Recently, a novel non-invasive method to calculate Myocardial Work (MW) showed a strong correlation with invasive work measurements.

Our aim was to investigate the ability of non-invasive MW indices to identify the ischaemic risk area during transient acute coronary occlusion (TACO).

### Methods and results

The study population comprises 50 individuals with critical coronary stenosis (CCS). Echocardiography recordings were obtained before coronary angiography, during TACO and after revascularization to measure global longitudinal strain (GLS), Myocardial Work Index (MWI), Myocardial Constructive Work (MCW), Myocardial Wasted work (MWW), Myocardial work efficiency (MWE).

Compared to baseline, we found a significant reduction of GLS (p = 0.005), MWI, MCW and MWE (p<0.001) during TACO.

### Conclusions

The non-invasive measurement of MW parameters is a sensitive and early marker of myocardial ischemia during TACO.

## Introduction

The prompt recognition of acutely ischemic myocardium has crucial therapeutic and prognostic implications [1].

**Funding:** This study was partly supported through an ESC Training Grant 2019, awarded to JS. There was no additional external funding received for this study.

**Competing interests:** The authors have declared that no competing interests exist.

With the introduction of more sensitive cardiac biomarkers, the fourth universal definition of myocardial infarction [2] has been released, taking into account myocardial injury detected by necrosis biomarkers, together with clinical symptoms, ECG changes or new regional wall motion abnormalities.

However, regional wall motion abnormalities deserve special considerations in this setting, as they appear early after flow reduction in the temporal sequence of the ischemic cascade [3]. The routine evaluation of myocardial function by echocardiography in the acute setting is mainly based on the visual assessment of wall motion. Such qualitative method has well recognized limitations [4,5] and may fail to distinguish subtle ischemia-induced signs in regional mechanics.

Recent studies showed that two-dimensional speckle tracking echocardiography (2D-STE) might identify an impairment of longitudinal function downward of critical coronary stenosis [6–13], differentiating acutely ischemic segments from both normal and dysfunctional myocardium [14].

Nevertheless, since strain parameters are load dependent, they might not reflect systolic function accurately [15,16] in specific settings.

Myocardial Work Index (MWI), a non-invasive method to quantify myocardial work using segmental strain and a standardized LV pressure (LVP) curve has been recently introduced [17–19].

Recently, Boe et al. [20] investigated the ability of regional MWI to identify acute coronary occlusion; however, their study was focused on patients with acute myocardial infarction (AMI).

Therefore, the aim of this study was to assess the impact of transient acute coronary occlusion on non-invasive myocardial work and 2D-STE-derived Longitudinal Strain (LS) to evaluate the impact of myocardial ischemia on these sensitive indices of LV function.

## Materials and methods

### Study population

We included 50 consecutive patients referred for coronary angiography in a single tertiary coronary care centre. Patients were included in the study population if they presented the following criteria:

- ≥18 years of age;

- clinical indication for coronary angiography;

- critical coronary stenosis (single vessel disease) as diagnosed during coronary angiography;

- gave their consent to participate.

Exclusion criteria were recent myocardial infarction (within 30 days), QRS-width of ≥120 ms, severe valvular disease, previous heart surgery, extensive comorbidity, or atrial fibrillation.

All patients were clinically and haemodynamically stable. The regional ethics committee (Comitato Etico Regione Calabria–Area Centro) approved the study, and all the patients provided written informed consent.

### Study timeline, procedures and analysis plan

Echocardiography recordings with simultaneous measurement of both non-invasive (NINV) automatic oscillometric and invasive intra-arterial blood pressure (INV) were obtained in the catheterization laboratory immediately before coronary angiography, during transient acute

coronary occlusion (TACO) and at the end of the procedure. The design of the study finds its parallel in studies previously performed by Edvardsen et al [21].

TACO was obtained by inflating a coronary balloon with a 1:1 balloon diameter-to-reference diameter ratio at the site of coronary stenosis at low pressure. Complete occlusion was verified injecting contrast medium, where TACO was defined as a TIMI flow of 0 (no perfusion) distal of the balloon inflation site. Pressure measurements and echocardiographic recordings during TACO were obtained starting at 60 seconds after balloon inflation.

The present study consists of the assessment of the impact of TACO on Myocardial Strain and Myocardial Work parameters. To this regard, Fig 1 depicts a representative example showing the changes in the strain-pressure loops from baseline, through TACO, to recover (after PCI).

## Echocardiographic analyses

Two-dimensional (2D) 4-chambers, 3-chambers and 2-chambers apical views were acquired, as previously described [13,22], with a frame rate ≥60 frames/s and, then, transferred to a dedicated workstation for the offline analysis (EchoPAC, GE Healthcare). The recordings were processed using an acoustic-tracking software (EchoPAC version 112.99, Research Release, GE Healthcare), which allowed an offline semi-automated analysis of speckle-based strain [23]. To calculate LV peak systolic longitudinal strain values and Post-systolic Shortening Index (PSI), a line was traced along the LV endocardium's inner border in each of the three apical views, and a region of interest, between the endocardial and epicardial borders, was recognized by the EchoPAC software. The region of interest was, then, adjusted to ensure that the wall thickness was incorporated in the analysis, avoiding the pericardium and following myocardial motion, as recommended. Results of LV peak systolic longitudinal strain and PSI were then provided by the software and analysed by an 18-segment model. PSI was calculated according to the formula by Kulkuski et al (PSI = (peak systolic strain − end-systolic strain)/ peak systolic strain) [14].

The timing of mitral and aortic valve closure and opening were obtained for Myocardial Work estimation.

**Calculation of non-invasive myocardial work.** MWI was calculated as the area of the LV pressure-strain loop (GE-Healthcare). Along with segmental and global values for myocardial work, a set of additional indices are also measured:

- Myocardial Constructive work (MCW): work performed by a segment during shortening in systole adding negative work during lengthening in IVR;

- Myocardial Wasted work (MWW): negative work performed by a segment during lengthening in systole adding work performed during shortening in IVR;

- Myocardial work efficiency (MWE): constructive work divided by the sum of constructive and wasted work (0–100%).

**Evaluation of regional function during transient coronary occlusion.** The ischemic risk area (IRA), downward the coronary artery transient occluded, and the non-risk area (NRA), were selected taking into consideration the vascular distribution of each coronary vessel, as previously detailed by Kukulski et al [14]. In brief, during transient occlusion of the right coronary artery (RCA), we considered IRA segments the inferior mid- and basal segments in apical two-chamber view. Transient occlusion of coronary arteries for some minutes has been shown to be safe in different clinical context [24,25]. During transient occlusion of the left circumflex coronary artery (LCx), the lateral basal- and mid segments, imaged in the apical four chamber

## MWI and GLS before, during and after transient coronary artery occlusion (TACO)

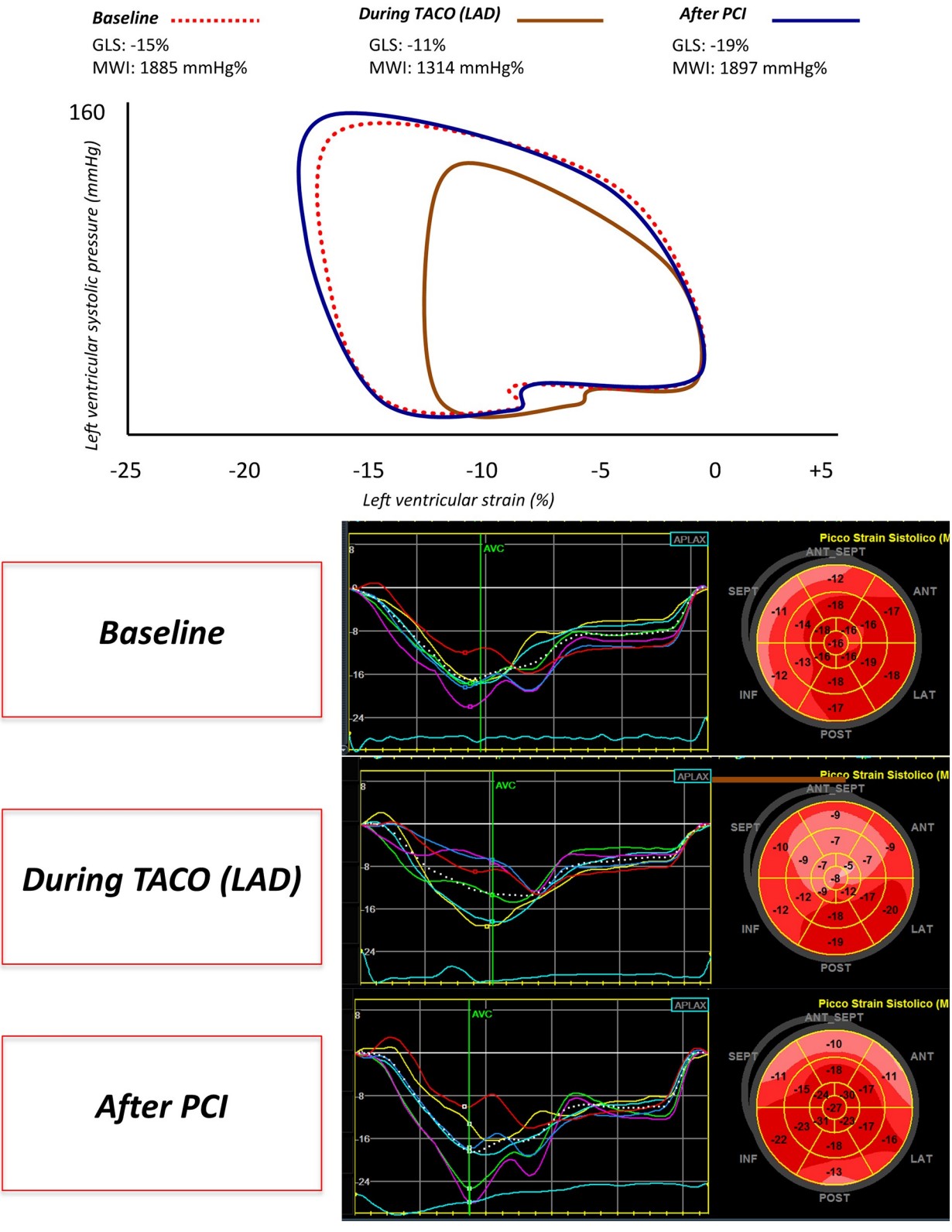

**Baseline** ....................
GLS: -15%
MWI: 1885 mmHg%

**During TACO (LAD)** ________
GLS: -11%
MWI: 1314 mmHg%

**After PCI** ________
GLS: -19%
MWI: 1897 mmHg%

**Fig 1. Strain-pressure loop and GLS changes during coronary occlusion.** Example of changes in Myocardial Work Index (MWI) and global longitudinal strain (GLS) at baseline, during transient coronary occlusion and after PCI. GLS: Global longitudinal strain. MWI: Myocardial work index. LAD: Left anterior descendent coronary artery. TACO: Transient acute coronary occlusion. PCI: Percutaneous coronary intervention.

view, were selected as IRA [14]. Both the septal mid- and apical segments in apical four-chamber view, were considered "at-risk" in presence of a transient occlusion of the left anterior descending coronary artery (LAD), as previously earlier described [11,14,21].

**Reproducibility study.** Fifteen echocardiographic examinations were randomly selected to assess inter-rater and intra-rater variability. Two operators performed the echocardiographic assessment in a blinded fashion. In addition, one of the two operators had to analyze the same series of exams twice without knowing it. Inter- and intra-rater reliability was then assessed using the intraclass correlation coefficient (ICC).

## Data analysis and statistics

Values are expressed as mean+SD or absolute numbers and/or percentages (%). Differences between groups were analysed with ANOVA. Comparison of continuous variables before versus during TACO were performed using the Wilcoxon U Test.

The intraclass correlation coefficient (ICC) was used to assess inter- and intra-rater reliability, as previously described [26].

Receiver operating characteristic (ROC) analysis was used to evaluate the diagnostic performance for each parameter. Significance of single ROC curves was assessed using the Hanley & McNeill method.

The statistical analyses were performed using SPSS v.21 (SPSS Inc., Chicago, IL, USA). A two-tailed P-value of 0.05 was considered significant.

## Results

### Study population

The study population comprises 50 individuals undergoing coronary angiography for clinical indication. All the patients included in the analysis had coronary artery disease (CAD) documented at invasive coronary angiography. Among CAD patients, seven were excluded for poor image quality, one patient was excluded from analysis for ongoing ventricular bigeminism during cardiac catheterization and one was excluded from the regional analysis for inadequate tracking. Patient characteristics, medication, and risk factors are listed in Table 1.

### Impact of acute coronary occlusion on global longitudinal strain and myocardial work indices

We found an impairment of LV systolic function during TACO, as demonstrated by reduced EF, GLS and MWI values. Six out 41 patients experienced chest pain above 6 (in a 1-to-10 analogic scale) during TACO; of those, 5 patients have undergone transient occlusion of the LAD and 1 of the RCA.

The average peak systolic GLS was significantly impaired during ischemia (Fig 1) compared to baseline (p = 0.005), with a significant reduction and a return to baseline values after reperfusion (p<0.001) (Fig 2). PSI was significantly increased during TACO (25.0±7.2%) compared to baseline (4.9±4%) (p<0.001) with a return to baseline values after PCI. Global MWI was significantly reduced during TACO compared to baseline (p<0.001). Similarly, global Myocardial Work Efficiency (MWE) index was significantly reduced during TACO (p<0.001), with a full recovery after PCI (p<0.001).

**Table 1. Baseline patients characteristics.**

| | PATIENTS (n = 41) |
|---|---|
| **Age (years)** | 67 ± 9 |
| **Male, n (%)** | 32 (78) |
| **Smoker, n (%)** | 11 (26) |
| **Hypercolesterolemia, n (%)** | 30 (73) |
| **Diabetes Mellitus, n (%)** | 11 (26) |
| **Hypertension, n (%)** | 32 (78) |
| **EF (%)** | 55±4 |
| **GLS (%)** | -17,2 ± 4,2 |
| **Target Vessel, n (%)** | |
| - LAD | 25 (60) |
| - LCx | 5 (12) |
| - RCA | 11 (26) |

In line with this last finding, we observed a significant increase of the global Myocardial Wasted Work index (MWW, p = 0.030) along with a significant reduction of the global Myocardial Constructive Work index (MCW, p<0.001) during TACO (Table 2).

As concern results from segmental analyses, regional MWE, measured acutely within the IRA, underlying the target vessel during transient coronary occlusion, was significantly decreased by 10% (p<0.001), compared with the NRA of the same patients. Also, regional LS (p<0.001) and regional PSI, measured acutely within the IRA, were observed being significantly reduced the former, and significantly increased the latter, compared with the NRA (-12.0±5.6% VS -16±4.8%, p<0.001; 18.4±13.3% VS 9.0±10.9%, p = 0.001) (Fig 3).

The diagnostic performance of those regional parameters measured acutely within the IRA, to ascertain the occurring of a transient acute coronary occlusion, were evaluated via ROC analyses. The area under the curve (AUC) was higher for regional MWE (AUC = 0.835, p<0.001) compared to both regional PSI (AUC = 0.792, p<0.001) and regional LS (AUC = 0.803, p<0.001) (Table 3).

## Blood pressure during the study

Mean non-invasive blood pressure (BP) values are reported in Table 2. Invasive BP from a study subgroup are shown in Table 4.

The average NINV systolic (SBP) values at baseline, during TACO and post-PCI were 143.8 ±17 mmHg, 123.6±23 mmHg and 139.8±21 mmHg, respectively (Table 2), with a significant reduction during ischemia compared to baseline (p<0.001) and a return to baseline values after reperfusion (p<0.001).

Similar to non invasive pressure values, average INV SBP values at baseline, during TACO and post-PCI were 141.2±23 mmHg, 118.2±24 mmHg and 140.1±21mmHg, respectively (Table 4), with values obtained under ischemia being significantly lower compared to baseline (p<0.001).

There were no significant differences between NINV and INV systolic and diastolic blood pressure measurements at baseline (p = 0.68 and p = 0.06), during TACO (p = 0.48 and p = 0.35) and post-PCI (p = 0.92 and p = 0.89), respectively.

INV showed good correlation with NINV systolic (ρ = 0.904; p<0.001) and diastolic (ρ = 0.684; p<0.001) blood pressure measurements at baseline. The correlation between INV and NINV was maintained under TACO (SBP: ρ = 0.756; p<0.001 and DBP: ρ = 0.808; p<0.001) and post PCI (SBP: ρ = 0.980; p<0.001 and DBP: ρ = 0.993; p<0.001).

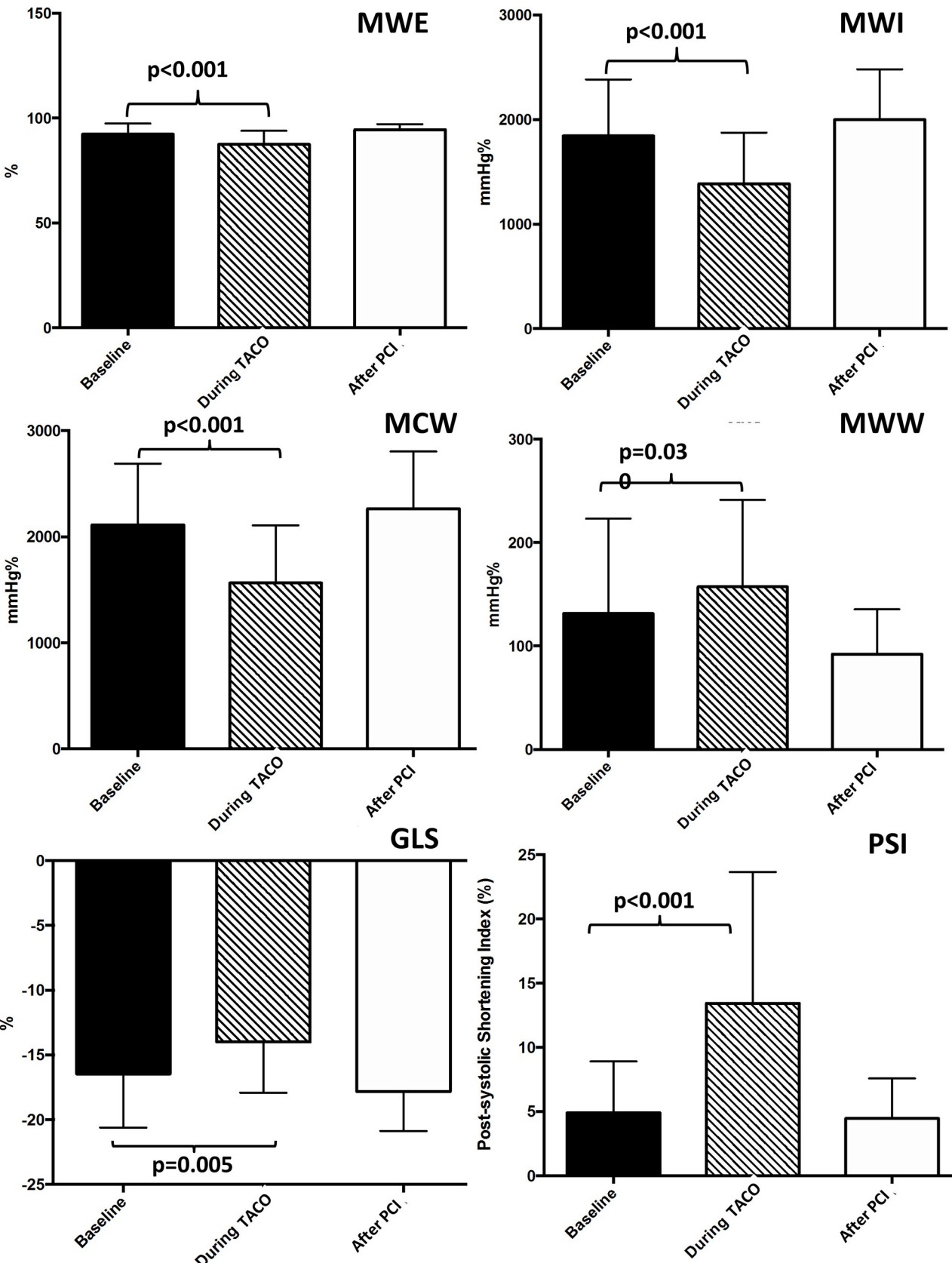

**Fig 2. Myocardial work indices and global longitudinal strain during transient acute coronary occlusion.** The graph shows the impact of acute coronary occlusion on myocardial work indices and global longitudinal strain. GLS: Global longitudinal strain. TACO: Transient acute coronary occlusion. PCI: Percutaneous coronary intervention. MWI: Myocardial work index. MCW: Myocardial Constructive work. MWW: Myocardial Wasted work. MWE: Myocardial work efficiency.

Finally, we demonstrated a strong correlation (R = 0.94, 95% CI = 0.75–0.96 p = 0.0001) between MWI calculated using INV vs NINV blood pressures by means of Bland-Altman and Youden plots (Fig 4).

## Reproducibility analyses

Intraclass correlation coefficient was very good for global MWI (ICC = 0.977; 95% CI: 0.944–0.991).

## Discussion

The main findings of our study are: 1) short and transient coronary occlusion results in an early reduction of non-invasive global myocardial work indices, as MWI, MCW, MWE, and in an increase of MWW; 2) MWI calculated using INV presents a strong correlation with MWI assessed using NINV blood pressures.

### Impairment of myocardial function during acute ischemia

An acute reduction in coronary blood flow induces a regional contractile dysfunction within a few seconds, resulting in impairment of regional deformation indices [27].

During ischemia, the longitudinal and circumferential systolic shortening of the ischemic segment are decreased, together with the radial thickening.

Moreover, diastolic relaxation is markedly impaired during ischemia and, in turn, the physiologic early diastolic thinning and lengthening are substituted by ongoing post-systolic thickening and shortening [28].

Consistent changes in early diastolic deformation have been demonstrated in several studies and proposed as an early marker of regional ischemia [14,29]. In our study, indeed, deformation indices were impaired during coronary occlusion in patients with chronic coronary syndrome. Indeed, not only the average peak systolic GLS was significantly impaired during ischemia, but also, not surprisingly, PSI was significantly increased during TACO compared to baseline, as consequence of the increased amount of post-systolic thickening occurred in the ischemic myocardium.

**Table 2. Echocardiographic data.**

|  | Baseline (n = 41) | During TACO (n = 41) | After PCI (n = 41) |
|---|---|---|---|
| **LVEF (%)** | 55±11 | 52±14 | 59±7 |
| **GLS (%)** | -16.5±4.1 | -14.0±3.9[a] | -17.8±3.0 |
| **SBP** | 143.8±17 | 123.6±23[a] | 139.8±21 |
| **DBP** | 81.2±9 | 74.8±15 | 76.2±13 |
| **SBP x Strain** | -2372±69 | -1730±90 | -2488±63 |
| **MWI (mmHg%)** | 1843±540 | 1387±488[a] | 2000±480 |
| **MWE (%)** | 92.3±5.1 | 87.5±6.3[a] | 94.5±2.5 |
| **MCW (mmHg%)** | 2112±577 | 1578±444[a] | 2220±455 |
| **MWW (mmHg%)** | 131.6±91.5 | 157.3±84.1[a] | 92.3±43.1 |

[a] = p<0.05 compared to baseline.

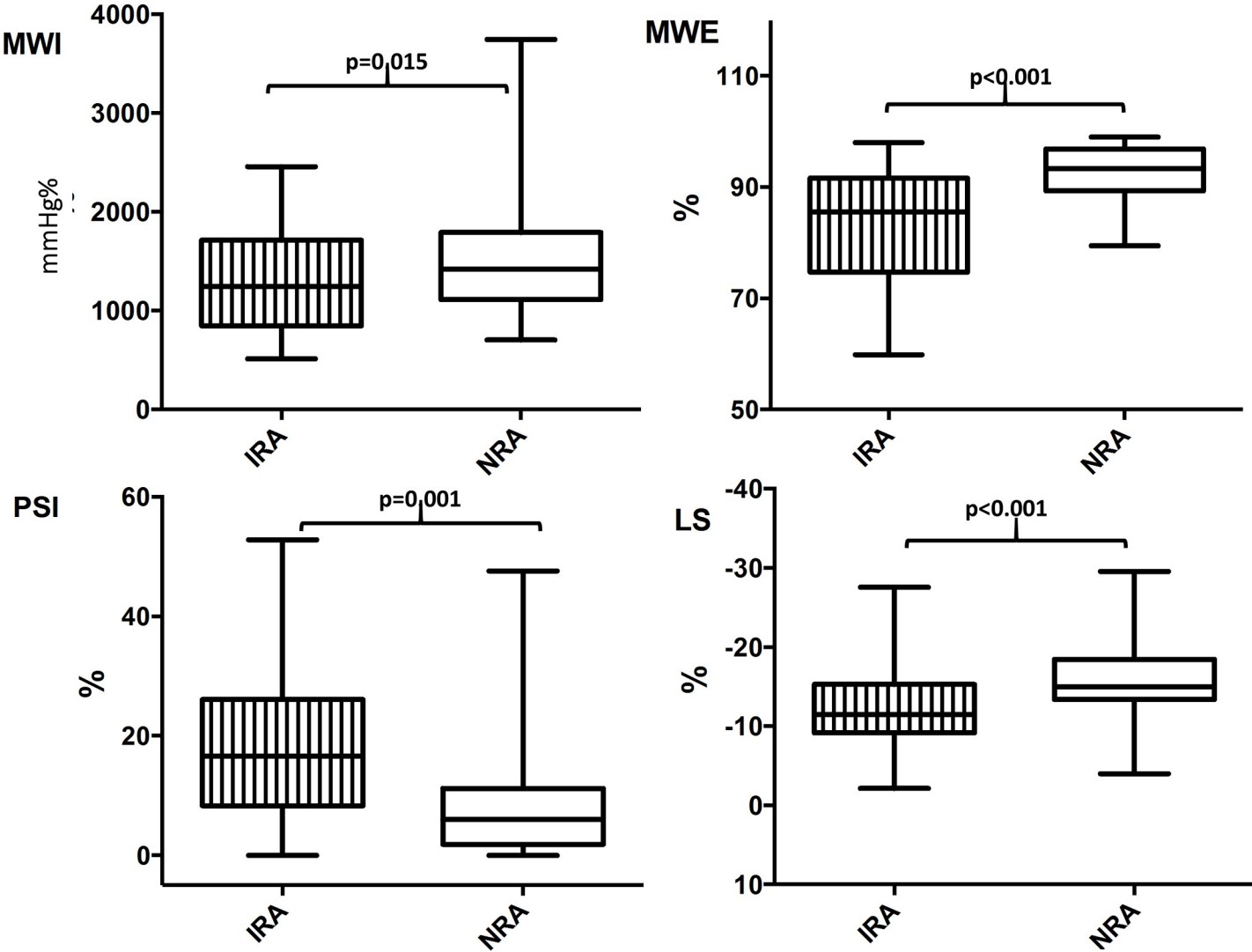

**Fig 3. Regional myocardial work and longitudinal strain parameters during transient acute coronary occlusion.** The chart shows regional MWE, MWI, LS and PSI, measured within the IRA (underlying the target vessel undergoing TACO), and compared to the NRA of the same patients. IRA = Ischemic risk area. NRA = Non-risk area. MWE = Myocardial work efficiency. MWI = Myocardial work index. PSI = Post systolic shortening index. LS = Longitudinal strain.

In 1987, Lazar et al [30] demonstrated a significant decrease in stroke work index, assessed with invasive measurements of LV Pressure-Volume loops, in 17 dogs when the proximal left anterior descending coronary artery was occluded for 45 minutes.

**Table 3. AUCs calculated by ROC curves for regional MWE, MWI, LS, PSI.**

|  | AUC | 95% C.I. | p |
|---|---|---|---|
| **MWE** | 0.835 | 0.74–0.94 | <0.001 |
| **MWI** | 0.766 | 0.67–0.87 | <0.001 |
| **LS** | 0.803 | 0.71–0.90 | <0.001 |
| **PSI** | 0.792 | 0.68–0.90 | <0.001 |

AUC = Area under the curve. ROC = Receiver operating characteristic. MWE = Myocardial work efficiency.

MWI = Myocardial work index. PSI = Post systolic shortening index. LS = Longitudinal strain.

**Table 4. Invasive pressures.**

| Invasive pressures | Baseline | During TACO | After PCI |
|---|---|---|---|
| SBP | 141.2±23 | 118.2±24 [a] | 140.1±21 |
| DBP | 76.1±10 | 70.5±16 | 75.7±12 |

[a] = p<0.05 compared to baseline.

We assessed for the first time in humans the impact of transient acute coronary occlusion, obtained by 60 minutes of balloon inflation, on a novel index that non-invasively estimate the myocardial work.

In agreement with findings by Lazar et al [30], obtained by means of invasive measurements in a dog model, we have observed that non-invasive MWI was significantly reduced during TACO compared to baseline values (p<0.001). Similarly, its derived indices as MCW and MWE index were significantly reduced during TACO (p<0.001).

## Myocardial work and ventricular function

Global and regional LV function is dependent on sub-endocardial blood flow [31]. Indices of longitudinal deformation can be influenced by the loading conditions, which limits their accuracy. Invasively measured Myocardial Work was introduced as marker of ventricular contractility since the 1970s [32–35]. It was later shown to provide similar physiological information to pressure/strain loops [36].

More recently, Russell et al [17] introduced a method for calculating non-invasive MW, on the basis of speckle tracking analysis with the estimation of LV pressure from brachial artery cuff pressure. The NORRE sub-study provided reference ranges for non-invasive MW, reporting a good reproducibility [37].

Most recently, Chan et al. [38] reported results of MW indices in three cardiovascular conditions, such as hypertension, ischaemic and not-ischaemic dilated cardiomyopathy. Particularly, they demonstrated a high impact of blood pressure on MW indices and a significant increase of MWI in hypertensive patients compared to controls, despite normal global longitudinal strain values.

In this regard, since myocardial work indices encompass multiple hemodynamic factors, they may—at least in part—complement and correct the estimation of systolic function compared to the sole strain measurements.

## Application of non-invasive myocardial work in clinical practice

Our findings might have a relevant impact on clinical practice. The identification and quantitation of non-invasive myocardial work abnormalities could serve as a valuable adjunct to the conventional diagnostic approach to chest pain patients, as they might be helpful to recognize early LV impairment in a more sensitive fashion, allowing the quicker identification of myocardial ischemia, with relevant impact both on patient's prognosis and clinical management workflow [39,40]. Furthermore, since data acquisition is totally non-invasive and safe and we have demonstrated MWI can be alternatively calculated both with INV or NINV, this novel method is appropriate for the monitoring of myocardial function even at short intervals after coronary revascularization. Early diagnosis or exclusion of critical coronary stenoses might be very useful both for the clinical management and for the prognostic impact [40]. In this regard, it should be pointed out that the echocardiographic assessment was performed in the catheterization laboratory on patients laying on the interventional table real-time during the procedure. This suggests that a rapid assessment to check for residual ischemia after initial PCI is ultimately

**A**

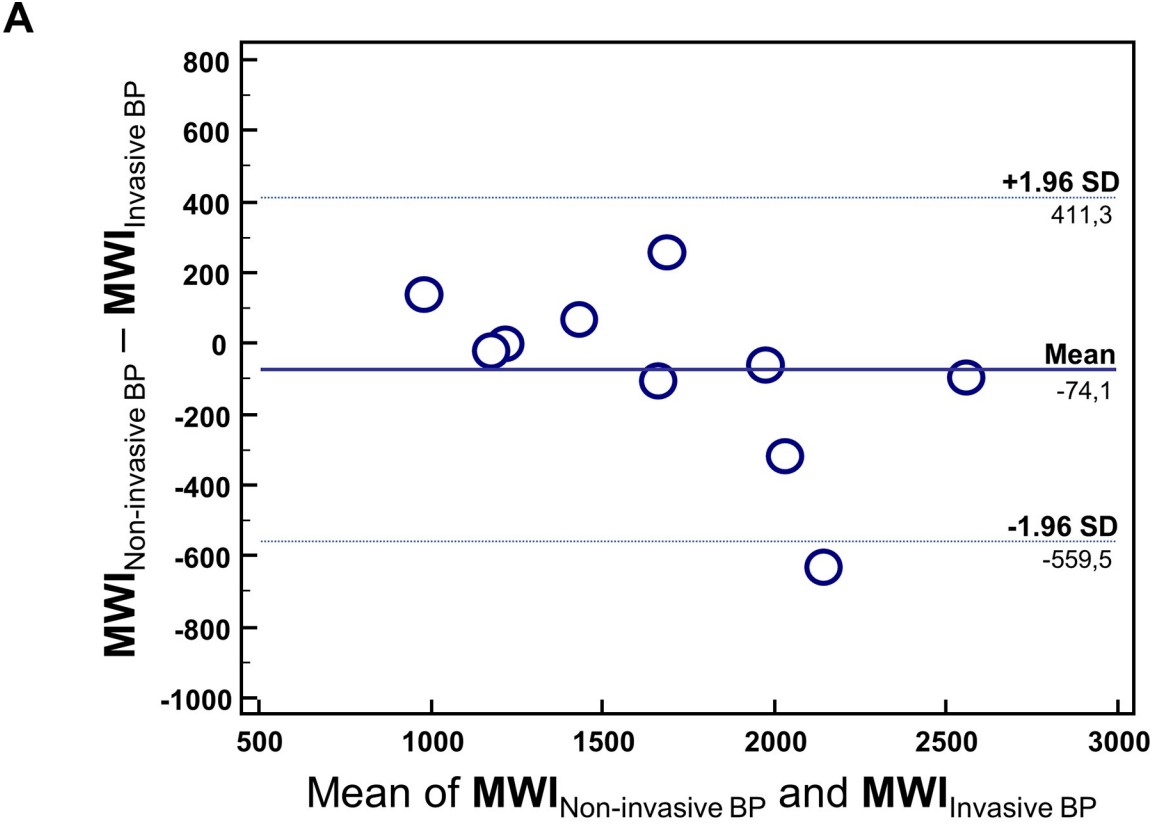

**B**

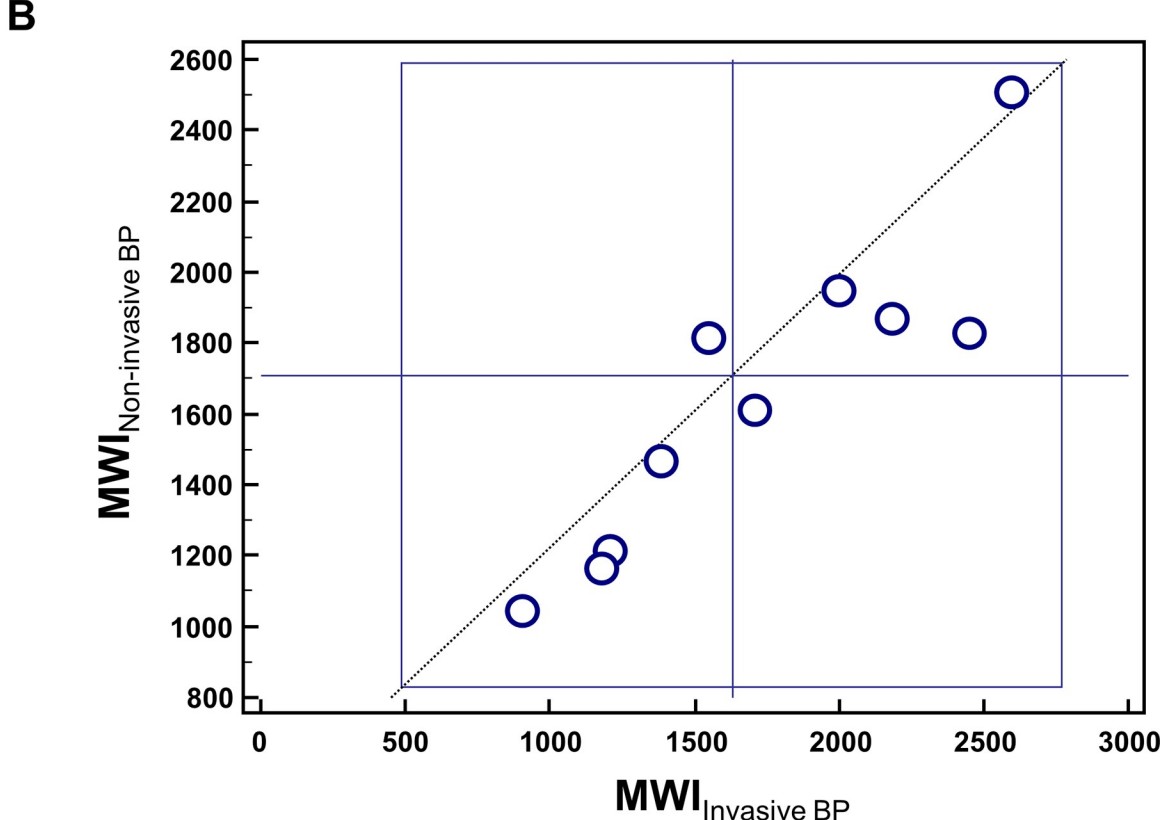

**Fig 4. Invasive vs non-invasive blood pressure measurement.** (A) Bland-Altman Plot comparing MWI obtained using invasive BP versus non invasive BP. (B) Youden Plot showing the scatter between MWI obtained using invasive BP versus non invasive BP.

possible. In this regard, the prognostic relevance of a timely complete coronary revascularization has been repeatedly demonstrated [41–43]. Finally, the new non-invasive MWI might be a more sensitive and precise alternative to visual assessment to distinguish between segments that benefited and did not benefit from PCI. In this regard, our results confirm previous evidence on global noninvasive work parameters and extends those results to regional LV assessment [44].

## Limitations

The design of the study with transient coronary occlusion may represent a study limitation, as the coronary occlusion has generally a longer duration in patients with acute myocardial infarction. However, the study design has also its strengths due to the controlled setting where each patient is his/her own control.

Echocardiographic exams in 16% of patients were not analysable due to insufficient image quality. Although these results are in line with recent studies [37], they could limit the applicability of our findings. MW analysis with contrast agents might improve its feasibility.

## Conclusions

Assessment of non-invasive Myocardial Work Indices is able to detect an impairment of LV function very early during coronary occlusion.

## Author Contributions

**Data curation:** Isabella Leo, Alberto Polimeni.

**Funding acquisition:** Jolanda Sabatino, Ciro Indolfi.

**Investigation:** Jolanda Sabatino, Isabella Leo, Carmen Spaccarotella, Annalisa Mongiardo.

**Methodology:** Jolanda Sabatino, Giovanni Di Salvo.

**Software:** Sabato Sorrentino.

**Supervision:** Salvatore De Rosa, Giovanni Di Salvo, Ciro Indolfi.

**Validation:** Giovanni Di Salvo.

**Visualization:** Salvatore De Rosa, Carmen Spaccarotella, Annalisa Mongiardo, Sabato Sorrentino.

**Writing – original draft:** Jolanda Sabatino.

**Writing – review & editing:** Salvatore De Rosa, Giovanni Di Salvo, Ciro Indolfi.

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
