## [Decision Letter · Decision Letter 0]

27 Jul 2020

PONE-D-20-18937

Non-invasive myocardial work is reduced during transient acute coronary occlusion.

PLOS ONE

Dear Dr. De Rosa,

Thank you for submitting your manuscript to PLOS ONE. After careful consideration, we feel that it has merit but does not fully meet PLOS ONE’s publication criteria as it currently stands. Therefore, we invite you to submit a revised version of the manuscript that addresses the points raised during the review process.

ACADEMIC EDITOR: All issues raised by expert reviewers are required.

We look forward to receiving your revised manuscript.

Kind regards,

Vincenzo Lionetti, M.D., PhD

Academic Editor

PLOS ONE

Journal Requirements:

2.  Thank you for including your ethics statement:  "The ethics review board approved the study, and all the patients provided written informed consent.".   

i) Please amend your current ethics statement to include the full name of the ethics committee/institutional review board(s) that approved your specific study.

ii) Once you have amended this/these statement(s) in the Methods section of the manuscript, please add the same text to the “Ethics Statement” field of the submission form (via “Edit Submission”).

"This study was partly supported through an ESC Training Grant 2019, awarded to Jolanda Ssabatino. The funders had no role in study design, data collection and analysis, decision to publish, or preparation of the manuscript".

i) Please provide an amended statement that declares *all* the funding or sources of support (whether external or internal to your organization) received during this study, as detailed online in our guide for authors at http://journals.plos.org/plosone/s/submit-now.  Please also include the statement “There was no additional external funding received for this study.” in your updated Funding Statement.

ii) Please include your amended Funding Statement within your cover letter. We will change the online submission form on your behalf.

Reviewers' comments:

Reviewer's Responses to Questions

**Comments to the Author**

1. Is the manuscript technically sound, and do the data support the conclusions?

Reviewer #1: Yes

Reviewer #2: Yes

2. Has the statistical analysis been performed appropriately and rigorously? 

Reviewer #1: Yes

Reviewer #2: Yes

3. Have the authors made all data underlying the findings in their manuscript fully available?

Reviewer #1: Yes

Reviewer #2: Yes

4. Is the manuscript presented in an intelligible fashion and written in standard English?

Reviewer #1: Yes

Reviewer #2: Yes

5. Review Comments to the Author

Reviewer #1: In everyday clinical practice the afterload dependency of all means of systolic ventricular function is still a challenge and unfortunately also often ignored. The noninvasive estimation of myocardial work invented by Prof Smiseth and his group in Oslo has moved the field a significant step forward as the afterload can be accounted for in the noninvasive work estimation to measure myocardial function. The method is now clinical available and the authors should be applauded in their work to validate this method. Thus, this paper by Sabation et al increases the knowledge in the field and well written.

Major comments:

• Why is no data on regional dysfunction included? I would have been very interesting to see the data from the different regions during the transient occlusion of the corresponding artery and probably increase the sensitivity of the method.

• The authors should use their data to demonstrate the possible increased or additional value of myocardial work vs strain.

Minor comments:

• In fact, the design of the study finds its parallel in some studies performed by Edvardsen and Skulstad in the early days of strain validation (for instance JACC, 2001). This should be addressed in the methods and included in the reference list.

• The authors use the term acute coronary occlusion and the abbreviation ACO. This confused this reviewer as none of the patients presented acute coronary occlusion, but critical coronary stenosis (Methods line 79). I suggest that this abbreviation should be solely used according to clinically acute occlusion as a thrombotic event like in an acute infarction. Why not stick to the term “transient acute coronary occlusion” as used in the heading also in the text and the figures?

• How many of the patients expressed chest pain during the angioplasty? This may have impact on their finding and should be discussed.

• The data in the supplementary tables should preferably be included in the main manuscript, either in the text or as tables.

• Please use SI units and replace “sec” by “s”.

• I suggest simplifying page 6 line 136 to “Two operators performed….”

• The design of the study with transient coronary occlusion should be discussed as a limitation as the occlusion has a longer duration in patients with acute infarction. However, the design also has its strengths due to the controlled situation where each patient is his/her own control.

• Include “n=41” also in table 2?

• Why is the term PTCA and not PCI used in the figures ? Replace ?

Reviewer #2: The manuscript treats an interesting topic, has a good structure, and is well written.

Major issues:

• Methods: if the presence of a critical stenosis is an inclusion criterion and every patient signed a written consent, it is licit to presume that coronary angiography and PCI (with the study-related measurements) happened at different times; how can a "planned coronary angiography" be an inclusion criterion as well? please clarify.

• the "Data analysis and statistics" section needs revision and expansion.

Minor issues:

• Potential clinical applications of an innovative diagnostic method is crucial and authors opportunely added an apposite section; nevertheless if the idea is that the method could be potentially useful in the setting of ACS and have a "relevant clinical impact both on patient's prognosis and clinical management" because more sensitive and able to detect functional abnormalities earlier than standard echographic methods, all these statements need to be supported by evidences in the literature. No references for the potential timesaving of the method, for the impact on prognosis of earlier diagnosis, for the sensitivity comparison among echographic techniques are present in the section. Please add some references (if already cited in the introduction, some can be re-cited).

• Data analysis and statistics: some parameters are also expressed as percentages (see table 1).

• Few language errors might be corrected; e.g.:

1. line 75: probably the past tense for "present" would be more correct

2. line 293: "eventually" is probably not the most appropriate word

6. PLOS authors have the option to publish the peer review history of their article (what does this mean?). If published, this will include your full peer review and any attached files.

Reviewer #1: **Yes: **Helge Skulstad

Reviewer #2: No

---

## [Author Response · Author response to Decision Letter 0]

14 Oct 2020

Editorial requirements:

Authors’ response: we revised the manuscript according to the journal’s style requirements.

2. Thank you for including your ethics statement: "The ethics review board approved the study, and all the patients provided written informed consent.". 

i) Please amend your current ethics statement to include the full name of the ethics committee/institutional review board(s) that approved your specific study.

ii) Once you have amended this/these statement(s) in the Methods section of the manuscript, please add the same text to the “Ethics Statement” field of the submission form (via “Edit Submission”).

Authors’ response: we amended the Ethics Statement as requested, adding the full name of the Ethics Review Board both in the revised manuscript and in the “Ethics Statement field” of the online submission form. 

"This study was partly supported through an ESC Training Grant 2019, awarded to Jolanda Sabatino. The funders had no role in study design, data collection and analysis, decision to publish, or preparation of the manuscript".

i) Please provide an amended statement that declares *all* the funding or sources of support (whether external or internal to your organization) received during this study, as detailed online in our guide for authors at http://journals.plos.org/plosone/s/submit-now. Please also include the statement “There was no additional external funding received for this study.” in your updated Funding Statement.

ii) Please include your amended Funding Statement within your cover letter. We will change the online submission form on your behalf. 

Authors’ response: we amended the Funding statement as requested in the revised manuscript.

Authors’ response: please notice that we have no Supporting Information files anymore, as the only files we had in the original version of the manuscript were added to the main content upon request by the external reviewers.

Response to Reviewers' comments:

5. Review Comments to the Author

Reviewer #1: In everyday clinical practice the afterload dependency of all means of systolic ventricular function is still a challenge and unfortunately also often ignored. The noninvasive estimation of myocardial work invented by Prof Smiseth and his group in Oslo has moved the field a significant step forward as the afterload can be accounted for in the noninvasive work estimation to measure myocardial function. The method is now clinical available and the authors should be applauded in their work to validate this method. Thus, this paper by Sabatino et al increases the knowledge in the field and well written.

Authors’ response: we thank the Reviewer for the general good comment about our manuscript.

Major comments:

• Why is no data on regional dysfunction included? I would have been very interesting to see the data from the different regions during the transient occlusion of the corresponding artery and probably increase the sensitivity of the method.

Authors’ response: we thank the Reviewer for this useful suggestion. Accordingly, we have now included in the manuscript results on regional dysfunction of myocardial work parameters, obtained acutely during transient coronary occlusion of the corresponding artery (page 6 and 9, lines 136-148 and 196-201, Fig 3). 

• The authors should use their data to demonstrate the possible increased or additional value of myocardial work vs strain.

Authors’ response: we thank the Reviewer for this observation. The diagnostic performance of those regional parameters measured acutely within the IRA, to ascertain the occurring of a transient acute coronary occlusion, were evaluated via ROC analysis. The area under the curve (AUC) was higher for regional MWE (AUC=0.835, p<0.001) compared to both regional PSI (AUC=0.792, p<0.001) and regional LS (AUC=0.803, p<0.001) (Table 3). 

We added these results and a table (Table 3) in our manuscript (page 9, lines 202-206). 

Minor comments:

• In fact, the design of the study finds its parallel in some studies performed by Edvardsen and Skulstad in the early days of strain validation (for instance JACC, 2001). This should be addressed in the methods and included in the reference list.

Authors’ response: we thank the Reviewer and referenced this study (ref. 21) as correctly suggested.

• The authors use the term acute coronary occlusion and the abbreviation ACO. This confused this reviewer as none of the patients presented acute coronary occlusion, but critical coronary stenosis (Methods line 79). I suggest that this abbreviation should be solely used according to clinically acute occlusion as a thrombotic event like in an acute infarction. Why not stick to the term “transient acute coronary occlusion” as used in the heading also in the text and the figures?

Authors’ response: we thank the Reviewer and replaced the wording “acute coronary occlusion” (AC0) with “transient acute coronary occlusion” (TACO) throughout the manuscript.

• How many of the patients expressed chest pain during the angioplasty? This may have impact on their finding and should be discussed.

Authors’ response: we thank the Reviewer and we added these data in Results Section (page 9, lines 183-185). “Six out 41 patients experienced chest pain over 6 (in a 1-to-10 analogic scale) during TACO; of those, 5 patients have undergone transient occlusion of the LAD and 1 of the RCA vessel.”

• The data in the supplementary tables should preferably be included in the main manuscript, either in the text or as tables.

Authors’ response: according to the Reviewer’s suggestion Supplemental Table 1 is now Table 4 and the content of Supplemental Table 2 has been included in the text of the main manuscript.

• Please use SI units and replace “sec” by “s”.

Authors’ response: we replaced “sec” by “s.

• I suggest simplifying page 6 line 136 to “Two operators performed….”

Authors’ response: we simplified the sentence as requested.

• The design of the study with transient coronary occlusion should be discussed as a limitation as the occlusion has a longer duration in patients with acute infarction. However, the design also has its strengths due to the controlled situation where each patient is his/her own control.

Authors’ response: we thank the Reviewer for the suggestion, and we included this argument in the Study Limitation Section (page 16, lines 332-335).

• Include “n=41” also in table 2?

Authors’ response: we included “n=41” also in table 2, as requested.

• Why is the term PTCA and not PCI used in the figures? Replace?

Authors’ response: we replaced PTCA with PCI.

Reviewer #2: The manuscript treats an interesting topic, has a good structure, and is well written.

Major issues:

• Methods: if the presence of a critical stenosis is an inclusion criterion and every patient signed a written consent, it is licit to presume that coronary angiography and PCI (with the study-related measurements) happened at different times; how can a "planned coronary angiography" be an inclusion criterion as well? please clarify.

Authors’ response: we apologize for the lack of clarity here. The sentence has been revised (page 4) to better reflect what was really meant:

“-clinical indication for coronary angiography;”.

• the "Data analysis and statistics" section needs revision and expansion.

Authors’ response: we thank the Reviewer for this suggestion. We reviewed and expanded the "Data analysis and statistics" Section, as requested. 

Minor issues:

• Potential clinical applications of an innovative diagnostic method is crucial and authors opportunely added an apposite section; nevertheless if the idea is that the method could be potentially useful in the setting of ACS and have a "relevant clinical impact both on patient's prognosis and clinical management" because more sensitive and able to detect functional abnormalities earlier than standard echographic methods, all these statements need to be supported by evidences in the literature. No references for the potential timesaving of the method, for the impact on prognosis of earlier diagnosis, for the sensitivity comparison among echographic techniques are present in the section. Please add some references (if already cited in the introduction, some can be re-cited).

Authors’ response: we thank the Reviewer for this comment. As suggested we revised the paragraph on potential clinical applications and provided the necessary supporting evidence (page 16, lines 311-330):

“Our findings might have a relevant impact on clinical practice. The identification and quantitation of non-invasive myocardial work abnormalities could serve as a valuable adjunct to the conventional diagnostic approach to chest pain patients, as they might be helpful to recognize early LV impairment in a more sensitive fashion, allowing the quicker identification of myocardial ischemia, with relevant impact both on patient’s prognosis and clinical management workflow (39-40). Furthermore, since data acquisition is totally non-invasive and safe and we have demonstrated MWI can be alternatively calculated both with INV or NINV, this novel method is appropriate for the monitoring of myocardial function even at short intervals after coronary revascularization. Early diagnosis or exclusion of critical coronary stenoses might be very useful both for the clinical management and for the prognostic impact (40). In this regard, it should be pointed out that the echocardiographic assessment was performed in the catheterization laboratory on patients laying on the interventional table real-time during the procedure. This suggests that a rapid assessment to check for residual ischemia after initial PCI is ultimately possible. In this regard, the prognostic relevance of a timely complete coronary revascularization has been repeatedly demonstrated (41-43). Finally, the new non-invasive MWI might be a more sensitive and precise alternative to visual assessment to distinguish between segments that benefited and did not benefit from PCI. In this regard, our results confirm previous evidence on global non-invasive work parameters and extends those results to regional LV assessment (44).”

• Data analysis and statistics: some parameters are also expressed as percentages (see table 1).

Authors’ response: we thank the Reviewer and corrected it as requested.

• Few language errors might be corrected; e.g.:

1. line 75: probably the past tense for "present" would be more correct

2. line 293: "eventually" is probably not the most appropriate word

Authors’ response: we apologise for the errors and corrected them as suggested.

---

## [Decision Letter · Decision Letter 1]

25 Nov 2020

PONE-D-20-18937R1

Non-invasive myocardial work is reduced during transient acute coronary occlusion.

PLOS ONE

Dear Dr. De Rosa,

Thank you for submitting your manuscript to PLOS ONE. After careful consideration, we feel that it has merit but does not fully meet PLOS ONE’s publication criteria as it currently stands. Therefore, we invite you to submit a revised version of the manuscript that addresses the points raised during the review process.

ACADEMIC EDITOR: All issues raised by expert reviewers are required in order to support the conclusions.

We look forward to receiving your revised manuscript.

Kind regards,

Vincenzo Lionetti, M.D., PhD

Academic Editor

PLOS ONE

Reviewers' comments:

Reviewer's Responses to Questions

**Comments to the Author**

1. If the authors have adequately addressed your comments raised in a previous round of review and you feel that this manuscript is now acceptable for publication, you may indicate that here to bypass the “Comments to the Author” section, enter your conflict of interest statement in the “Confidential to Editor” section, and submit your "Accept" recommendation.

Reviewer #1: (No Response)

Reviewer #2: (No Response)

2. Is the manuscript technically sound, and do the data support the conclusions?

Reviewer #1: No

Reviewer #2: (No Response)

3. Has the statistical analysis been performed appropriately and rigorously? 

Reviewer #1: I Don't Know

Reviewer #2: (No Response)

4. Have the authors made all data underlying the findings in their manuscript fully available?

Reviewer #1: No

Reviewer #2: (No Response)

5. Is the manuscript presented in an intelligible fashion and written in standard English?

Reviewer #1: No

Reviewer #2: (No Response)

6. Review Comments to the Author

Reviewer #1: No new version is added, only two old versions.

I guess the authors have done something wrong. They should receive help from the editorial office.

Reviewer #2: The version PONE-D-20-18937_R1_reviewer contains two identical versions of the manuscript without any of the changes the authors have described in their responses. This issue was communicate to the editor in a previous email. If it is my fault I do apologize in advance. At the present state of things my review cannot be completed.

7. PLOS authors have the option to publish the peer review history of their article (what does this mean?). If published, this will include your full peer review and any attached files.

Reviewer #1: No

Reviewer #2: No

---

## [Author Response · Author response to Decision Letter 1]

27 Nov 2020

Editorial requirements:

Authors’ response: we revised the manuscript according to the journal’s style requirements.

2. Thank you for including your ethics statement: "The ethics review board approved the study, and all the patients provided written informed consent.". 

i) Please amend your current ethics statement to include the full name of the ethics committee/institutional review board(s) that approved your specific study.

ii) Once you have amended this/these statement(s) in the Methods section of the manuscript, please add the same text to the “Ethics Statement” field of the submission form (via “Edit Submission”).

Authors’ response: we amended the Ethics Statement as requested, adding the full name of the Ethics Review Board both in the revised manuscript and in the “Ethics Statement field” of the online submission form. 

"This study was partly supported through an ESC Training Grant 2019, awarded to Jolanda Sabatino. The funders had no role in study design, data collection and analysis, decision to publish, or preparation of the manuscript".

i) Please provide an amended statement that declares *all* the funding or sources of support (whether external or internal to your organization) received during this study, as detailed online in our guide for authors at http://journals.plos.org/plosone/s/submit-now. Please also include the statement “There was no additional external funding received for this study.” in your updated Funding Statement.

ii) Please include your amended Funding Statement within your cover letter. We will change the online submission form on your behalf. 

Authors’ response: we amended the Funding statement as requested in the revised manuscript.

Authors’ response: please notice that we have no Supporting Information files anymore, as the only files we had in the original version of the manuscript were added to the main content upon request by the external reviewers.

Original Response to Reviewers' comments:

5. Review Comments to the Author

Reviewer #1: In everyday clinical practice the afterload dependency of all means of systolic ventricular function is still a challenge and unfortunately also often ignored. The noninvasive estimation of myocardial work invented by Prof Smiseth and his group in Oslo has moved the field a significant step forward as the afterload can be accounted for in the noninvasive work estimation to measure myocardial function. The method is now clinical available and the authors should be applauded in their work to validate this method. Thus, this paper by Sabatino et al increases the knowledge in the field and well written.

Authors’ response: we thank the Reviewer for the general good comment about our manuscript.

Major comments:

• Why is no data on regional dysfunction included? I would have been very interesting to see the data from the different regions during the transient occlusion of the corresponding artery and probably increase the sensitivity of the method.

Authors’ response: we thank the Reviewer for this useful suggestion. Accordingly, we have now included in the manuscript results on regional dysfunction of myocardial work parameters, obtained acutely during transient coronary occlusion of the corresponding artery (page 6 and 9, lines 136-148 and 196-201, Fig 3). 

• The authors should use their data to demonstrate the possible increased or additional value of myocardial work vs strain.

Authors’ response: we thank the Reviewer for this observation. The diagnostic performance of those regional parameters measured acutely within the IRA, to ascertain the occurring of a transient acute coronary occlusion, were evaluated via ROC analysis. The area under the curve (AUC) was higher for regional MWE (AUC=0.835, p<0.001) compared to both regional PSI (AUC=0.792, p<0.001) and regional LS (AUC=0.803, p<0.001) (Table 3). 

We added these results and a table (Table 3) in our manuscript (page 9, lines 202-206). 

Minor comments:

• In fact, the design of the study finds its parallel in some studies performed by Edvardsen and Skulstad in the early days of strain validation (for instance JACC, 2001). This should be addressed in the methods and included in the reference list.

Authors’ response: we thank the Reviewer and referenced this study (ref. 21) as correctly suggested.

• The authors use the term acute coronary occlusion and the abbreviation ACO. This confused this reviewer as none of the patients presented acute coronary occlusion, but critical coronary stenosis (Methods line 79). I suggest that this abbreviation should be solely used according to clinically acute occlusion as a thrombotic event like in an acute infarction. Why not stick to the term “transient acute coronary occlusion” as used in the heading also in the text and the figures?

Authors’ response: we thank the Reviewer and replaced the wording “acute coronary occlusion” (AC0) with “transient acute coronary occlusion” (TACO) throughout the manuscript.

• How many of the patients expressed chest pain during the angioplasty? This may have impact on their finding and should be discussed.

Authors’ response: we thank the Reviewer and we added these data in Results Section (page 9, lines 183-185). “Six out 41 patients experienced chest pain over 6 (in a 1-to-10 analogic scale) during TACO; of those, 5 patients have undergone transient occlusion of the LAD and 1 of the RCA vessel.”

• The data in the supplementary tables should preferably be included in the main manuscript, either in the text or as tables.

Authors’ response: according to the Reviewer’s suggestion Supplemental Table 1 is now Table 4 and the content of Supplemental Table 2 has been included in the text of the main manuscript.

• Please use SI units and replace “sec” by “s”.

Authors’ response: we replaced “sec” by “s.

• I suggest simplifying page 6 line 136 to “Two operators performed….”

Authors’ response: we simplified the sentence as requested.

• The design of the study with transient coronary occlusion should be discussed as a limitation as the occlusion has a longer duration in patients with acute infarction. However, the design also has its strengths due to the controlled situation where each patient is his/her own control.

Authors’ response: we thank the Reviewer for the suggestion, and we included this argument in the Study Limitation Section (page 16, lines 332-335).

• Include “n=41” also in table 2?

Authors’ response: we included “n=41” also in table 2, as requested.

• Why is the term PTCA and not PCI used in the figures? Replace?

Authors’ response: we replaced PTCA with PCI.

Reviewer #2: The manuscript treats an interesting topic, has a good structure, and is well written.

Major issues:

• Methods: if the presence of a critical stenosis is an inclusion criterion and every patient signed a written consent, it is licit to presume that coronary angiography and PCI (with the study-related measurements) happened at different times; how can a "planned coronary angiography" be an inclusion criterion as well? please clarify.

Authors’ response: we apologize for the lack of clarity here. The sentence has been revised (page 4) to better reflect what was really meant:

“-clinical indication for coronary angiography;”.

• the "Data analysis and statistics" section needs revision and expansion.

Authors’ response: we thank the Reviewer for this suggestion. We reviewed and expanded the "Data analysis and statistics" Section, as requested. 

Minor issues:

• Potential clinical applications of an innovative diagnostic method is crucial and authors opportunely added an apposite section; nevertheless if the idea is that the method could be potentially useful in the setting of ACS and have a "relevant clinical impact both on patient's prognosis and clinical management" because more sensitive and able to detect functional abnormalities earlier than standard echographic methods, all these statements need to be supported by evidences in the literature. No references for the potential timesaving of the method, for the impact on prognosis of earlier diagnosis, for the sensitivity comparison among echographic techniques are present in the section. Please add some references (if already cited in the introduction, some can be re-cited).

Authors’ response: we thank the Reviewer for this comment. As suggested we revised the paragraph on potential clinical applications and provided the necessary supporting evidence (page 16, lines 311-330):

“Our findings might have a relevant impact on clinical practice. The identification and quantitation of non-invasive myocardial work abnormalities could serve as a valuable adjunct to the conventional diagnostic approach to chest pain patients, as they might be helpful to recognize early LV impairment in a more sensitive fashion, allowing the quicker identification of myocardial ischemia, with relevant impact both on patient’s prognosis and clinical management workflow (39-40). Furthermore, since data acquisition is totally non-invasive and safe and we have demonstrated MWI can be alternatively calculated both with INV or NINV, this novel method is appropriate for the monitoring of myocardial function even at short intervals after coronary revascularization. Early diagnosis or exclusion of critical coronary stenoses might be very useful both for the clinical management and for the prognostic impact (40). In this regard, it should be pointed out that the echocardiographic assessment was performed in the catheterization laboratory on patients laying on the interventional table real-time during the procedure. This suggests that a rapid assessment to check for residual ischemia after initial PCI is ultimately possible. In this regard, the prognostic relevance of a timely complete coronary revascularization has been repeatedly demonstrated (41-43). Finally, the new non-invasive MWI might be a more sensitive and precise alternative to visual assessment to distinguish between segments that benefited and did not benefit from PCI. In this regard, our results confirm previous evidence on global non-invasive work parameters and extends those results to regional LV assessment (44).”

• Data analysis and statistics: some parameters are also expressed as percentages (see table 1).

Authors’ response: we thank the Reviewer and corrected it as requested.

• Few language errors might be corrected; e.g.:

1. line 75: probably the past tense for "present" would be more correct

2. line 293: "eventually" is probably not the most appropriate word

Authors’ response: we apologise for the errors and corrected them as suggested.

Response to the additional Reviewers' comments:

Reviewer #1: No new version is added, only two old versions.

I guess the authors have done something wrong. They should receive help from the editorial office.

Authors’ response: we apologise for the mishap. We are sorry for the time you lost around this revision. We are not able to explain what was the issue why you didn’t receive the revised manuscript file. All files have been now newly uploaded and the previous ones deleted from the editorial manager to avoid any further issues.

Reviewer #2: The version PONE-D-20-18937_R1_reviewer contains two identical versions of the manuscript without any of the changes the authors have described in their responses. This issue was communicate to the editor in a previous email. If it is my fault I do apologize in advance. At the present state of things my review cannot be completed.

Authors’ response: we apologise for the mishap, which we were aware of only upon receipt of the decision. We are sorry for the time you lost around this revision. We are not able to explain what was the issue why you didn’t receive the revised manuscript file. All files have been now newly uploaded and the previous ones deleted from the editorial manager to avoid any further issues.

---

## [Decision Letter · Decision Letter 2]

9 Dec 2020

Non-invasive myocardial work is reduced during transient acute coronary occlusion.

PONE-D-20-18937R2

Dear Dr. De Rosa,

We’re pleased to inform you that your manuscript has been judged scientifically suitable for publication and will be formally accepted for publication once it meets all outstanding technical requirements.

Kind regards,

Vincenzo Lionetti, M.D., PhD

Academic Editor

PLOS ONE

Additional Editor Comments (optional):

Reviewers' comments:

Reviewer's Responses to Questions

**Comments to the Author**

1. If the authors have adequately addressed your comments raised in a previous round of review and you feel that this manuscript is now acceptable for publication, you may indicate that here to bypass the “Comments to the Author” section, enter your conflict of interest statement in the “Confidential to Editor” section, and submit your "Accept" recommendation.

Reviewer #1: All comments have been addressed

2. Is the manuscript technically sound, and do the data support the conclusions?

Reviewer #1: Yes

3. Has the statistical analysis been performed appropriately and rigorously? 

Reviewer #1: Yes

4. Have the authors made all data underlying the findings in their manuscript fully available?

Reviewer #1: Yes

5. Is the manuscript presented in an intelligible fashion and written in standard English?

Reviewer #1: Yes

6. Review Comments to the Author

Reviewer #1: No further comments. The paper fulfill the criteria needed for publication. They have addressed all the comment properly.

7. PLOS authors have the option to publish the peer review history of their article (what does this mean?). If published, this will include your full peer review and any attached files.

Reviewer #1: **Yes: **Helge Skulstad

---

## [Editor Report · Acceptance letter]

15 Dec 2020

PONE-D-20-18937R2 

Non-invasive myocardial work is reduced during transient acute coronary occlusion. 

Dear Dr. De Rosa:

I'm pleased to inform you that your manuscript has been deemed suitable for publication in PLOS ONE. Congratulations! Your manuscript is now with our production department. 

Kind regards, 

on behalf of

Prof. Vincenzo Lionetti 

Academic Editor

PLOS ONE